# Double burden of malnutrition in Nepal: A trend analysis of protein-energy malnutrition and High Body Mass Index using the data from Global Burden of Disease 2010–2019

**Priza Pradhananga**[1]*, **Archana Shrestha**[1,2,3], **Nabin Adhikari**[1], **Namuna Shrestha**[1], **Mukesh Adhikari**[3,4], **Nicole Ide**[5], **Saurya Dhungel**[6], **Swornim Bajracharya**[3], **Anu Aryal**[1,3,7]

1 Department of Public Health and Community Programs, Dhulikhel Hospital- Kathmandu University Hospital, Dhulikhel, Nepal, 2 Department of Chronic Disease Epidemiology, Yale School of Public, New Haven, Nepal, 3 Institute for Implementation Science and Health, Kathmandu, Nepal, 4 Department of Health Policy and Management, Gillings School of Global Public Health, University of North Carolina, Chapel Hill, North Carolina, United States of America, 5 Resolve to Save Lives, Vital Strategies, New York, New York, United States of America, 6 Department of Epidemiology, University of Washington School of Public Health, Seattle, United States of America, 7 Department of Health Policy and Management, University of California Los Angeles, Los Angeles, California, United States of America

* prizapradhananga@gmail.com

## Abstract

### Background

The co-existence of undernutrition and overnutrition is a global public health threat. We aim to report the burden of both nutritional deficiency (Protein-Energy Malnutrition) and overweight (high Body Mass Index) in Nepal over a decade (2010–2019) and observe the changes through trend charts.

### Methods

We did a secondary data analysis using the Institute for Health Metrics and Evaluation (IHME)'s Global Burden of Disease (GBD) database to download age-standardized data on Protein Energy Malnutrition (PEM) and high Body Mass Index (BMI). We presented the trend of death, Disability Adjusted Life Years (DALYs), Years of Life Lost (YLL), and Years Lost due to Disability (YLD) of PEM and high BMI in Nepal from 2010 to 2019 and also compared data for 2019 among South Asian countries.

### Results

Between 2010 and 2019, in Nepal, the Disability Adjusted Life Years (DALYs) due to PEM were declining while high BMI was in increasing trend. Sex-specific trends revealed that females had higher DALYs for PEM than males. In contrast, males had higher DALYs for high BMI than females. In 2019, Nepal had the highest death rate for PEM (5.22 per 100,000 populations) than any other South Asian country. The burden of PEM in terms of DALY was higher in under-five children (912 per 100,000 populations) and elderly above 80 years old (808.9 per 100,000 populations), while the population aged 65–69 years had the

**Data Availability Statement:** Our study included use of aggregated third party data publicly available from the IHME GBD database. The Global Burden

of Disease data are freely available to download from the GBD results tool (https://ghdx.healthdata.org/gbd-results-tool). A user account is not needed. Researchers can select the required year, sex, age group, location, metrics, measures, causes, and risk factors of their interest. Once the selections have been made, the data can be downloaded in CSV file format by providing a valid email address. When the file is ready for download, the website sends an email with the link to download the data. The downloaded data can easily be opened in Excel for analysis or other statistical software.

**Funding:** The author(s) received no specific funding for this work.

**Competing interests:** The authors have declared that no competing interests exist.

highest burden of high BMI (5893 per 100,000 populations). In the last decade, the DALYs for risk factors contributing to PEM such as child growth failure (stunting and wasting), unsafe water, sanitation and handwashing, and sub-optimal breastfeeding have declined in Nepal. On the contrary, the DALYs for risk factors contributing to high BMI, such as a diet high in sugar-sweetened beverages, a diet high in trans fatty acid, and low physical activity, have increased. This could be a possible explanation for the increasing trend of high BMI and decreasing trend of PEM.

## Conclusion

Rapidly growing prevalence of high BMI and the persistent existence of undernutrition indicate the double burden of malnutrition in Nepal. Public health initiatives should be planned to address this problem.

## Introduction

The co-existence of undernutrition and overnutrition is a global public health threat. Adults with obesity are at increased risk of developing non-communicable diseases such as diabetes, hypertension, stroke, cardiovascular diseases, and some forms of cancer [1]. On the other hand, being underweight is linked with consequences such as premature mortality, infirmities, impaired intellectual development, and poor self-rated health and well-being [2]. In 2016, 1.9 billion adults aged 18 years and above were overweight or obese, whereas 462 million adults were underweight globally [3]. Around 33% of the world's population suffers from at least one form of malnutrition like wasting, stunting, vitamin and mineral deficiencies, overweight, and obesity [4]. Along with the health impacts, malnutrition also has serious consequences on countries' social and economic development. Mortality and morbidities due to malnutrition could cost almost US$3.5 trillion annually, of which US$2.5 trillion is covered by undernutrition and micronutrient deficiency while US$1.4 trillion by overweight and obesity-related non-communicable diseases, which have a direct loss in human capital and productivity [5].

This double burden of malnutrition has been observed in many developing countries, including countries in South Asia, including Nepal [6–9]. In South Asian countries, the prevalence of underweight among children aged 24 to 59 months was 37%, 38%, 19%, 28%, and 29% in Bangladesh, India, Maldives, Nepal, and Pakistan, respectively. The prevalence of overweight amongst children was higher in Pakistan (7%) and Maldives(9%) and lower in Nepal, India, and Bangladesh (between 2% and 4%) [10, 11]. The Global Nutrition Report estimated that 17.4% of female and 16% of male adults in Nepal were underweight in 2016, while 22.8% of females and 19.1% of male adults were overweight in the same year [12]. The National Demographic and Health Survey (NDHS) 2016 reported the prevalence of underweight as 19.2% and the prevalence of overweight as 18.2% among Nepali adults [13].

Historically, significant focus in Nepal has been given to curbing undernutrition due to its high prevalence, and overnutrition is not given much attention. The NDHS, which is done every five years, started collecting overweight measures only since 2016; also the NDHS 2021 report has not been published yet, so there are not enough data points to observe a trend [14]. In this study, we used the Global Burden of Disease(GBD) database to analyze the trend of undernutrition and overnutrition in Nepal over a decade (2010–2019). We used Protein-Energy malnutrition, a primary form of malnutrition in Nepali children [15], as a measure of undernutrition and high Body Mass Index (BMI) as a measure of overnutrition. We further

analyzed the trend by gender and additionally compared the burden with other South Asian countries. The findings from this study will help to observe trends in both forms of malnutrition in Nepal and help policymakers develop comprehensive nutrition strategies in Nepal that address the double burden of malnutrition.

## Methods

### Data source

We did a secondary data analysis using the Institute for Health Metrics and Evaluation's (IHME) Global Burden of Diseases (GBD) database available online through the GBD results tool. The GBD contains estimates of the burden of diseases, including incidence, mortality, prevalence, years of life lost due to premature mortality(YLL), years lived with disability (YLD), and disability-adjusted life years (DALY) of various illnesses and injuries for 195 countries. GBD defines high BMI in adults 20 years and older as BMI greater than 20–25 kg/m2 and in children 19 years and younger as being obese or obesity as per International Obesity Task Force standards. They define PEM as a health loss associated with moderate and severe acute wasting [16, 17]. The detailed description of metrics, data collection procedures, and analytical approaches used for GBD are reported elsewhere [18]. The data input source tool of GBD showed a total of 9 high BMI-related research articles and 4 PEM-related articles used to create GBD estimates for Nepal [19].

### Analysis

We downloaded the estimates and their 95% confidence interval for age-standardized deaths, Disability-Adjusted Life Years (DALYs), Years of Life Lost (YLL), Years Lost to Disability (YLD), rates per 100,000 population for Protein Energy Malnutrition (PEM), and high Body Mass Index (BMI) for Nepal from 2010–2019 in.csv format. We repeated the same data extraction procedure for other South Asian countries as comparators, including India, Bangladesh, Bhutan, Pakistan, Sri Lanka, Maldives, and Afghanistan. We used Microsoft Excel to create tables and figures. The download data steps with our comparison analysis and charts are included in the S1 File.

In this article, we present the trend of death, DALYs, YLL, and YLD of PEM, and high BMI in Nepal from 2010–2019 and compare data of 2019 among South Asian countries. We also presented changes in DALYs due to various risk factors in Nepal from 2010 to 2019.

## Results

Fig 1 illustrates the trend in DALYs for PEM and high BMI in Nepal. Between 2010 to 2019, the DALYs due to PEM is declining, while the DALYs due to high BMI are increasing. The DALYs for PEM has dropped from 390 per 100,000 population in 2010 to 208 per 100,000 population in 2019. Meanwhile, the DALYs for High BMI has increased steadily from 953 in 2010 to 1354 per 100,000 population in 2019. Sex-specific trends revealed that females had higher DALY for PEM than males in the past decade. In contrast, males had higher DALYs for high BMI than females in the past decade.

Table 1 compares the death rate, DALYs, YLDs, and YLLs\ related to PEM and high BMI in Nepal with other South Asian countries. Although PEM was declining, Nepal had the highest death rate(5.22 per 100,000 population) for PEM than other South Asian countries. DALY for PEM in Nepal was slightly lower than the South Asia average but higher than Bangladesh, Bhutan, Sri Lanka, and the Maldives. With high BMI, Nepal had a higher death rate and DALY than Bangladesh and Maldives but lower than Bhutan, India, Pakistan, Sri Lanka, and

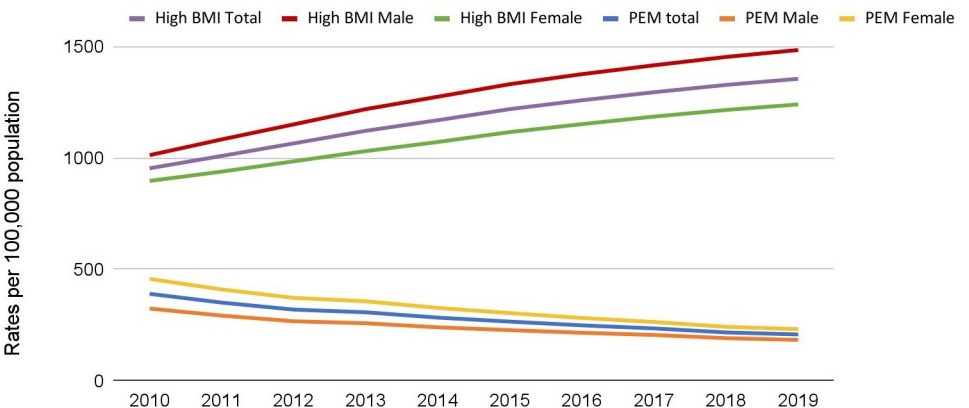

**Fig 1. Trend of DALY for Protein Energy Malnutrition and High Body Mass Index in Nepal (2010–2019).**

Afghanistan. Among the South Asian countries, Pakistan had the highest DALYs (287.44 per 100,000) due to PEM, while Afghanistan had the highest death (177.28 per 100,000) and DALYs (5098.61 per 100,000) due to high BMI.

Fig 2 presents the age-wise distribution of DALYs for PEM and high BMI in Nepal in 2019, respectively. The burden of PEM in terms of DALYs was higher in under-five children and elderly above 80 years old while lowest in 10–14 years old (40.6 per 100,000 population). The rate of PEM started to decrease in the population above five years but began to increase consistently after the age of 50. In the case of high BMI, children below 19 years had a low burden, while the population aged 65–69 had the highest burden (5893 per 100,000 population). The rate of high BMI is observed to increase with the increase in age.

In Fig 3, we noticed a pattern of various contributing risk factors over the past decade in Nepal. Over the past decade, the DALYs for child growth failure (stunting and wasting), unsafe

**Table 1. DALY, YLL, YLD, Death for PEM and High BMI among South Asian countries (2019).**

| Location | Protein Energy Malnutrition (PEM) rates per 100,000 population (95% CI) | | | | High Body Mass Index (BMI) rates per 100,000 population (95% CI) | | | |
|---|---|---|---|---|---|---|---|---|
| | Death | DALY | YLD | YLL | Death | DALY | YLD | YLL |
| South Asia | 1.68 (2.21–1.23) | 217.60 (275.54–167.94) | 106.91 (150.39–68.02) | 110.68 (150.02–77.87) | 52.0 (77.93–30.92) | 1769.15 (2440.57–1043.51) | 373.15 (571.24–215.54) | 1356.58 (1945.68–812.62) |
| Nepal | 5.22 (6.86–3.89) | 208.01 (274.82–154.06) | 45.94 (66.65–29.22) | 162.06 (228.40–114.11) | 42.14 (70.53–20.02) | 1354.33 (2144.73–707.98) | 310.15 (497.09–157.17) | 1044.18 (1689.75–510.56) |
| Bangladesh | 2.03 (2.70–1.42) | 139.97 (193.56–98.33) | 46.17 (65.07–29.04) | 93.79 (144.50–55.63) | 33.29 (56.83–15.18) | 1149.48 (1836.28–590.90) | 233.99 (389.63–117.07) | 915.49 (1489.42–452.11) |
| Bhutan | 0.51 (1.23–0.18) | 83.61 (141.93–50.51) | 46.83 (70.61–28.17) | 36.77 (94.71–12.86) | 55.74 (87.03–27.83) | 1733.59 (2601.51–972.05) | 385.60 (597.06–217.41) | 1347.98 (2088.36–726.66) |
| India | 1.18 (1.77–0.77) | 208.04 (274.31–153.14) | 125.37 (176.17–79.73) | 82.67 (127.20–52.79) | 51.59 (75.68–30.12) | 1697.83 (2370.73–1015.12) | 383.16 (585.03–223.20) | 1314.66 (1909.49–784.11) |
| Pakistan | 3.50 (4.84–2.40) | 287.44 (388.80–209.55) | 56.24 (79.84–36.34) | 231.20 (327.84–154.47) | 92.24 (141.10–51.29) | 2777.63 (4165.90–1605.49) | 447.23 (699.17–245.93) | 2330.39 (3548.34–1308.63) |
| Sri Lanka | 0.76 (1.01–0.57) | 142.87 (196.35–96.13) | 129.49 (183.10–81.61) | 13.38 (17.94–10.01) | 66.84 (104.80–36.77) | 2073.38 (3078.99–1210.16) | 659.88 (989.68–369.35) | 1413.49 (2195.84–791.38) |
| Afghanistan | 2.38 (3.61–1.57) | 202.04 (300.72–137.94) | 37.0 (52.28–23.27) | 165.04 (262.13–102.69) | 177.28(256.14–109.10) | 5098.61 (7244.70–3230.86) | 670.18 (985.84–413.34) | 4428.42 (6447.93–2732.61) |
| Maldives | 1.28 (1.61–0.98) | 185.89 (261.52–124.41) | 161.83 (237.78–101.59) | 24.06 (30.78–18.86) | 35.54 (58.66–17.96) | 1229.06 (1838.10–699.19) | 376.09 (591.59–204.21) | 852.97 (1307.06–467.49) |

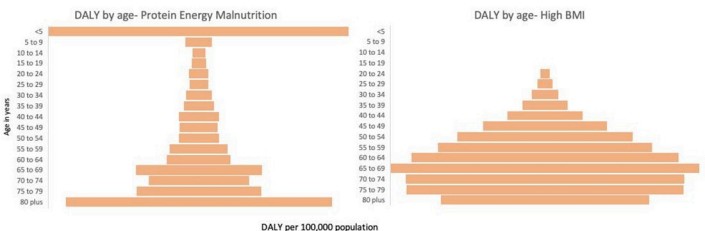

**Fig 2. Age-Wise distribution of DALY for PEM and High BMI in Nepal (2019).**

water, sanitation and handwashing, and sub-optimal breastfeeding, which are often associated with PEM, have declined. On the contrary, the DALYs for a diet high in sugar-sweetened beverages, a diet high in trans fatty acid, and low physical activity, which are often associated with high BMI, have increased.

## Discussion

Using the GBD data, we examined the trend of malnutrition (both protein-energy malnutrition and high body mass index) for the past ten years in Nepal. The findings of our study provide evidence for the existence of a dual burden of malnutrition. The pattern of undernutrition is declining; however, a significant population is still underweight. Meanwhile, overweight and obesity are becoming dominant forms of malnutrition, indicating a shift in the nutritional trend.

Prior studies in South Asian countries show consistent findings of the double burden of malnutrition with our study. A Bangladeshi study showed that 30% of the adults were underweight, 18.9% overweight, and 4.6% obese [20, 21]. Likewise, in 2012–13, a study in Pakistan indicated 13% undernutrition and 25% overweight in women [22]. A study in India also suggested that the high prevalence of undernutrition coexisted with overweight and obesity [2].

Our results examined that the trend of undernutrition is declining in Nepal. This is in line with studies in India and Nepal that showed that underweight among adults decreased by almost 15% [2] and stunting in under-five children decreased by 18% in urban settings [23]. Between 2006 to 2016, the NDHS reported a declining stunting prevalence among Nepali children from 39% to 29% and wasting from 49% to 36%.

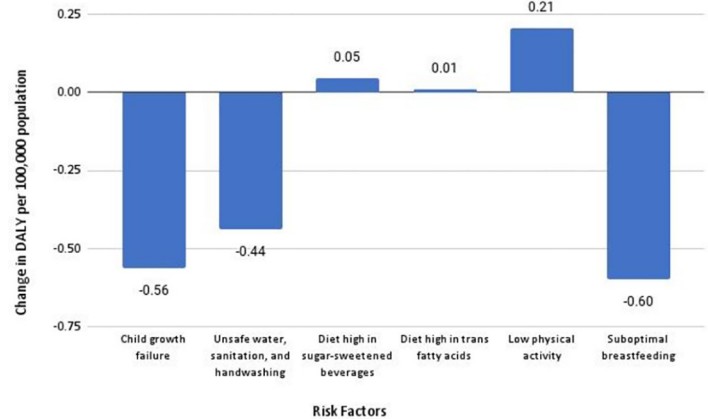

**Fig 3. Change in DALY due to various risk factors in Nepal between 2010–2019.**

From 2010 to 2019, we noticed a sharp rise in the rates of overweight and obesity in Nepal and other South Asian countries. India also showed an increasing pattern of obesity [24, 25]. From 1980 to 2013, the rate of obesity in South Asia increased by 5% [26]. Prevalence of obesity among women of reproductive age also increased by 6.2% in Bangladesh, 8.5% in Nepal, and 4.2% in India from 1996 to 2006 [27].

The findings of our study are also important as they highlight that women are at a higher risk of being undernourished than men. Gender disparity is observed in previous studies, showing that globally, women are more likely to be undernourished [28–30]. In the context of developing countries like Nepal, women's reproductive health, poverty, lack of education, low social status, low access to health and social services, household work patterns, gender disparities, and socio-economics disparities are the possible reasons for women's vulnerability towards suffering from malnourishment [31]. Though our study showed higher BMI in males than females, a few studies comparing obesity across countries show contrasting findings that the prevalence of obesity is typically higher in females [32–34].

We have found that the older population above 80 years was at more risk of being underweight. Possible explanations for this include aging is associated with loss of appetite, decrease in taste and smell, less physical activity, psychological disorders, deteriorating dental health causing difficulty in eating a variety of food, and chewing problems that can interfere with nutritional status resulting in malnourishment [35]. Our study also showed that under-five children were at higher risk of undernutrition. The lesser the age of children, the higher the risk of undernutrition [36–38]. Children under 12 months were twice as likely to be stunted than children aged 24 to 36 months [37]. Low household income, mother's educational status, antenatal checkup, the health-seeking practice of mothers, diarrhea and respiratory infections within one month of birth, and mother's nutritional status have been cited as the major associated factors for under-five malnutrition [38, 39].

Our study is the first to identify the trends in the dual burden of malnutrition among Nepalese over an extended period. We acknowledge that a large-scale primary data collection effort would have been the best approach to capture this trend in Nepal. In the absence of such, we have attempted this analysis through the Global Burden of Disease database. Our study is able to show the trend but is limited to offering explanations for such trends. Additional studies on various risk factors contributing to malnutrition in Nepal will be needed to identify areas of intervention.

## Policy implications

Our analysis has a few policy implications. First, observing the high burden of undernutrition, along with a rapidly increasing trend of DALYs associated with high BMI, the government and non-governmental key stakeholders should think of revising the existing policies to curb the double burden of nutrition in Nepal. Second, the increasing prevalence of overweight/ high BMI shows that either the current policies are ineffective or there is a lack of concrete policies and programs to support healthy lifestyle adoption for Nepali people. Since the risk for other chronic diseases and the subsequent costs associated with being overweight or obese are high, a low-income country like Nepal should timely intervene in this burgeoning problem. Third, with disparities associated with overweight and underweight being much starker by gender and age groups, there is a high need for targeted interventions. Fourth and more importantly, the concerned stakeholders should generate robust evidence on direct and indirect factors associated with overweight and underweight at federal, provincial, and local levels to effectively formulate policies to curb malnutrition at different levels.

## Conclusion

Our study provides evidence for the co-existence of undernutrition and overnutrition in Nepal. Nepal is going through a nutritional transition where undernutrition is declining but still prevalent, while obesity/overweight is increasing steadily. Undernutrition is higher in females, while high BMI is higher in males. The prevalence of overweight/obesity is low in under-five children, but undernutrition remains highest in that age group. The dual burden of malnutrition is alarming and should be taken into consideration. Public health interventions should be planned to emphasize a healthy diet and lifestyle. Our findings also substantiate the need for nutritional strategies that address the situation of dual burden of malnutrition in Nepal.

## Supporting information

**S1 File.**
(XLSX)

## Author Contributions

**Conceptualization:** Priza Pradhananga, Archana Shrestha, Nabin Adhikari, Namuna Shrestha, Mukesh Adhikari, Nicole Ide, Saurya Dhungel, Swornim Bajracharya, Anu Aryal.

**Data curation:** Priza Pradhananga, Nabin Adhikari, Anu Aryal.

**Formal analysis:** Priza Pradhananga, Archana Shrestha, Nabin Adhikari, Namuna Shrestha, Mukesh Adhikari, Saurya Dhungel, Anu Aryal.

**Investigation:** Archana Shrestha, Anu Aryal.

**Methodology:** Priza Pradhananga, Anu Aryal.

**Supervision:** Archana Shrestha, Anu Aryal.

**Writing – original draft:** Priza Pradhananga.

**Writing – review & editing:** Priza Pradhananga, Archana Shrestha, Nabin Adhikari, Namuna Shrestha, Mukesh Adhikari, Nicole Ide, Saurya Dhungel, Swornim Bajracharya, Anu Aryal.

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
