## [Decision Letter · Decision Letter 0]

24 Feb 2022

PONE-D-21-41047Double Burden of Malnutrition in Nepal: Analysis of Data from Global Burden of Disease for Trend in Protein-Energy Malnutrition and High Body Mass IndexPLOS ONE

Dear Dr. Pradhananga,

Thank you for submitting your manuscript to PLOS ONE. After careful consideration, we feel that it has merit but does not fully meet PLOS ONE’s publication criteria as it currently stands. Therefore, we invite you to submit a revised version of the manuscript that addresses the points raised during the review process.

We look forward to receiving your revised manuscript.

Kind regards,

Pranil Man Singh Pradhan, M.D.

Academic Editor

PLOS ONE

Journal Requirements:

Reviewers' comments:

Reviewer's Responses to Questions

**Comments to the Author**

1. Is the manuscript technically sound, and do the data support the conclusions?

Reviewer #1: Partly

Reviewer #2: No

2. Has the statistical analysis been performed appropriately and rigorously? 

Reviewer #1: I Don't Know

Reviewer #2: No

3. Have the authors made all data underlying the findings in their manuscript fully available?

Reviewer #1: No

Reviewer #2: No

4. Is the manuscript presented in an intelligible fashion and written in standard English?

Reviewer #1: Yes

Reviewer #2: Yes

5. Review Comments to the Author

Reviewer #1: PONE-D-21-41047: Double Burden of Malnutrition in Nepal: Analysis of Data from Global Burden of Disease for Trend in Protein-Energy Malnutrition and High Body Mass Index

This study is aimed to assess the coexistence of undernutrition and overnutrition among the Nepalese population over a period of 2010 to 2019 using data from the Institute for Health Metrics and Evaluation (IHME)’s Global Burden of Disease (GBD) database. Overall, this study found that undernutrition is exists but decreasing in trend, whilst overnutrition is rising gradually. The author(s) are addressing the most important public health issues in Nepal. I would recommend the following points which need to be improved before the manuscript is accepted for publication.

MAJOR COMMENTS

• The authors have not stated the sampled populations, inclusion criteria, and sampling strategy to select the desired number of samples from the GBD database. Therefore, sampling strategy and exclusion criteria are fundamentals of sampling strategy and need to be stated in the manuscript transparently and using complete and clear descriptions.

• Why author(s) did include all ages instead of using a certain target age group? That might have reflected the heterogeneity and large variation of data. In addition, physiologically there is more likely to be obese with an increase in age, whilst the undernutrition problem is more prevalent in under 5 children. Therefore, the findings of this study could be affected by this phenomenon.

• Overall, the method section needs to revise extensively. The methods section needs to be reported in a way that another research with access to the same database would be able to reproduce the (sub) sample used in this study. For instances, data extracting procedure, own sampling strategy for secondary data, description of study variables, and statistical analyses used for this study, etc.

• Since this study is not based on the multi-country analysis, why did the author(s) include the other south Asian country's data in the result? This is not in line with the study objectives.

• The entire paper could benefit from additional proofreading and grammatical correction to improve clarity and directness.

MINOR COMMENTS

Abstract

• Overall abstract section needs to be reformat like a background, methods, results, and conclusions.

• Replace the word “report” with “identify or examine”

• Description of results in the abstract is mostly subjective. It is recommended to include objective measures of the results in this section.

• I would recommended to write clear and concise conclusions with brief recommendation.

Background

• The authors start from the macro idea, contextualize, and even identify the problem. The section is finished with the justification and the main objective of the work, there are some concerns that have to be addressed.

• The authors should provide the tangible rationale of the study. Why does this study need to conduct? What is already known? What do new findings implicit? In the background, all these things need to be well addressed.

• In the sentence “This double burden of malnutrition has been observed in many developing countries including countries in South Asia, including Nepal [6–9]” of these citations, number 8 study is not conducted in South Asia. Please revise it.

Results

• In figure 2, along with the age-wise distribution of PEM and BMI, it would be better to present gender-wise too.

• Only subjective results are presented for table 3, it is recommended to write numbers too.

 

Discussion

• The overall discussion needs to be specific. It could be more concise by only mentioning of most important discussion.

References

Reformat all references with PLOS ONE referencing style and revise the references # 3, 9, 12, 14, 16,17,18, 20, 25, 32, 42 etc….

Reviewer #2: This is interesting paper that aims for trend analysis of PEM and BMI of Nepal using the data from global burden of disease. Although the research article made an early attempt on trend analysis, the content in the paper seems to be all over the place. The research paper appeared to explain a lot of stuff but data provided were limited. I think only way the research article could be made workable is by being more specified and detailed.

1. The basic guidelines of research paper submission need to be followed. There is no any line numbering. It would have been easier to give suggestion indicating the line number. Please check the submission guidelines

2. Title of the study and the content present in the paper do not match.

3. Methodology section is like nonexistent in the paper. There is no information about sample size, sampling strategy, inclusion and exclusion criteria of the study. There is no information on any predictor or outcome variable while discussion section is filled with various predictors of high BMI and PEM. How was the statistical analysis done? The questions like: How were the data extracted and what software was used for analysis, remain unanswered. Methodology need to be explained well enough to make the paper more transparent and meet the criteria of replicablility.

4. BMI and PEM are quantitative variable. They require objective measurement and still they are prone to various errors. BMI and PEM are age sensitive. It would be interesting if the researcher had explained specifically on how information on PEM and BMI were assessed in GBD studies. Also, please provide clear operational definition of PEM and BMI.

5. The discussion section in research paper is basically based on findings of the research study and then one can explain how they fit with existing research and theory. The discussion in this research study is completely out of context. The aim of the study is trend analysis while discussion section was all about factors associated with BMI and PEM. In the discussion section, defend the findings of your own study than explaining results of other studies. Target population in the study is too wide/ vague.

6. The referencing style does not follow the guidelines given by PLOS ONE. For instance, reference number 18 is about a research article while in the reference section only URL is copy pasted. Please recheck your reference section.

6. PLOS authors have the option to publish the peer review history of their article (what does this mean?). If published, this will include your full peer review and any attached files.

Reviewer #1: **Yes: **Dev Ram Sunuwar

Reviewer #2: No

---

## [Author Response · Author response to Decision Letter 0]

24 Apr 2022

Reviewer #1

MAJOR COMMENTS

1. The authors have not stated the sampled populations, inclusion criteria, and sampling strategy to select the desired number of samples from the GBD database. Therefore, sampling strategy and exclusion criteria are fundamentals of sampling strategy and need to be stated in the manuscript transparently and using complete and clear descriptions.

● Response: Thank you for this comment. This is a secondary data analysis using Global Burden of Disease database. To add clarity to the readers we have now divided the methods section into two subsections- ‘data source’, and ‘analysis’, and have included additional details. 

2. Why did the author(s) include all ages instead of using a certain target age group? That might have reflected the heterogeneity and large variation of data. In addition, physiologically there is more likely to be obese with an increase in age, whilst the undernutrition problem is more prevalent in under 5 children. Therefore, the findings of this study could be affected by this phenomenon.

● Response: We used age standardized rates for overall trend analysis. We understand age-wise variation is important in case of malnutrition and high BMI, and have presented age wise distribution of DALY for both in the figure 2.

3. Overall, the method section needs to revise extensively. The methods section needs to be reported in a way that another research with access to the same database would be able to reproduce the (sub) sample used in this study. For instances, data extracting procedure, own sampling strategy for secondary data, description of study variables, and statistical analyses used for this study, etc.

● Response: Based on your comment #1, and this comment, we have revised the structure of our methods section. We have provided the link to the website where the data was downloaded from, specified the indicators, year, country, and measures (rate per 100,000) which ensures reproducibility of our analysis. 

4. Since this study is not based on the multi-country analysis, why did the author(s) include the other south Asian country's data in the result? This is not in line with the study objectives.

● Response: We included results from other South Asian countries to compare where Nepal stands. We mentioned this in line 83 in the introduction section, and in line 104 in the methods section. 

5. The entire paper could benefit from additional proofreading and grammatical correction to improve clarity and directness.

● Response: Thank you for your comment. We have extensively copy edited the manuscript this time around. 

MINOR COMMENTS

Abstract

6. Overall abstract section needs to be reformatted like a background, methods, results, and conclusions.

● Response: Thank you for this helpful note. We have now added those subsections in the abstract. 

7. Replace the word “report” with “identify or examine”

● Response: We have edited it in the new version. 

8. Description of results in the abstract is mostly subjective. It is recommended to include objective measures of the results in this section.

● Response: We have added numbers in the results section in the current revised manuscript. 

9. I would recommended to write clear and concise conclusions with brief recommendation.

● Response: Thank you. We have edited our conclusion section. 

Background

The authors start from the macro idea, contextualize, and even identify the problem. The section is finished with the justification and the main objective of the work, there are some concerns that have to be addressed.

10. The authors should provide the tangible rationale of the study. Why does this study need to conduct? What is already known? What do new findings implicit? In the background, all these things need to be well addressed.

● Response: We revisited the introduction section and made minor changes. Overall we felt it does answer questions raised by the reviewer. 

11. In the sentence “This double burden of malnutrition has been observed in many developing countries including countries in South Asia, including Nepal [6–9]” of these citations, number 8 study is not conducted in South Asia. Please revise it.

● Response: We have revised the above mentioned reference.

Results

12. In figure 2, along with the age-wise distribution of PEM and BMI, it would be better to present gender-wise too.

● Response: We have presented gender wise differences in the trend in PEM and high BMI in figure 1 already. 

13. Only subjective results are presented for table 3, it is recommended to write numbers too.

● Response: We do not have a Table 3, but we assumed you meant figure 3. Based on your feedback we added values for each bar in the graph. 

Discussion

14. The overall discussion needs to be specific. It could be more concise by only mentioning of most important discussion.

● Response: We have extensively edited the discussion section in response to reviewers’ comments. 

References

15. Reformat all references with PLOS ONE referencing style and revise the references # 3, 9, 12, 14, 16,17,18, 20, 25, 32, 42 etc….

● Response: Thankyou for addressing it. We have extensively revised the references section.

—-------------------------------------------------------------------------------------------------------------------

Reviewer #2: 

1. The basic guidelines of research paper submission need to be followed. There is no any line numbering. It would have been easier to give suggestion indicating the line number. Please check the submission guidelines

● Response: We appreciate the note, and have added line numbers in the revised version of the manuscript. 

2. Title of the study and the content present in the paper do not match.

● Response: Based on your feedback we have edited the title in this revised version. 

3. Methodology section is like nonexistent in the paper. There is no information about sample size, sampling strategy, inclusion and exclusion criteria of the study. There is no information on any predictor or outcome variable while discussion section is filled with various predictors of high BMI and PEM. How was the statistical analysis done? The questions like: How were the data extracted and what software was used for analysis, remain unanswered. Methodology need to be explained well enough to make the paper more transparent and meet the criteria of replicablility.

● Response: Thank you for this comment. This is a secondary data analysis using Global Burden of Disease database. To add clarity to the readers we have now divided the methods section into two subsections- ‘data source’, and ‘analysis’, and have included additional details. In this revised version we have now provided the link of website where the data was downloaded from. We have specified the indicators, year, country, and measures (rate per 100,000) which ensures reproducibility of our analysis. 

4. BMI and PEM are quantitative variable. They require objective measurement and still they are prone to various errors. BMI and PEM are age sensitive. It would be interesting if the researcher had explained specifically on how information on PEM and BMI were assessed in GBD studies. Also, please provide clear operational definition of PEM and BMI.

● Response: We agree that BMI and PEM are age sensitive, and thank you for pointing that out. We have now edited the methods section and have specified methods for age groups (>=20 years, and <=19years) for BMI. 

About operational definition, unfortunately because it is a secondary data analysis using GBD data, we could only use the definition provided by the GBD. 

In lines 80-82 we have clarified that we used PEM and high BMI as proxy indicators for undernutrtion and overnutrition. “We used Protein-Energy malnutrition, a primary form of malnutrition in Nepali children[15] as a measure of undernutrition, and high body mass index (BMI) as a measure of overnutrition.”

5. The discussion section in research paper is basically based on findings of the research study and then one can explain how they fit with existing research and theory. The discussion in this research study is completely out of context. The aim of the study is trend analysis while discussion section was all about factors associated with BMI and PEM. In the discussion section, defend the findings of your own study than explaining results of other studies. Target population in the study is too wide/ vague.

● Response: We have extensively edited the discussion section in response to reviewers’ comments. 

6. The referencing style does not follow the guidelines given by PLOS ONE. For instance, reference number 18 is about a research article while in the reference section only URL is copy pasted. Please recheck your reference section.

● Response: Thank you for raising this. We have edited the references section in the current revised manuscript. We hope that we satisfyingly addressed them.

---

## [Decision Letter · Decision Letter 1]

27 Jun 2022

PONE-D-21-41047R1Double Burden of Malnutrition in Nepal: A trend analysis of Protein-Energy Malnutrition and High Body Mass Index using the data from Global Burden of Disease 2010-2019PLOS ONE

Dear Dr. Pradhananga,

Thank you for submitting your manuscript to PLOS ONE. After careful consideration, we feel that it has merit but does not fully meet PLOS ONE’s publication criteria as it currently stands. Therefore, we invite you to submit a revised version of the manuscript that addresses the points raised during the review process.

We look forward to receiving your revised manuscript.

Kind regards,

Pranil Man Singh Pradhan, M.D.

Academic Editor

PLOS ONE

Reviewers' comments:

Reviewer's Responses to Questions

**Comments to the Author**

1. If the authors have adequately addressed your comments raised in a previous round of review and you feel that this manuscript is now acceptable for publication, you may indicate that here to bypass the “Comments to the Author” section, enter your conflict of interest statement in the “Confidential to Editor” section, and submit your "Accept" recommendation.

Reviewer #1: (No Response)

Reviewer #2: (No Response)

2. Is the manuscript technically sound, and do the data support the conclusions?

Reviewer #1: Partly

Reviewer #2: Partly

3. Has the statistical analysis been performed appropriately and rigorously? 

Reviewer #1: I Don't Know

Reviewer #2: No

4. Have the authors made all data underlying the findings in their manuscript fully available?

Reviewer #1: No

Reviewer #2: Yes

5. Is the manuscript presented in an intelligible fashion and written in standard English?

Reviewer #1: Yes

Reviewer #2: Yes

6. Review Comments to the Author

Reviewer #1: Double Burden of Malnutrition in Nepal: A trend analysis of Protein-Energy Malnutrition and High Body Mass Index using the data from Global Burden of Disease 2010-2019

Manuscript ID: PONE-D-21-41047R1

There are major issues with this manuscript as at present stage which has to be addressed.

1. Methodology: Transparency and thoroughness are essential in reporting, since it is strongly advised to not leave readers in the shadows about the procedure. All we know that author(s) have used secondary data from GBD database. However, things is GBD database contains large data with various information. The concern is how did you sample the required data, and how did you perform all these analyses? Therefore, sampling strategy, robust statistical analysis, software used for data analysis needs to be reported in a way that another research with access to the same database would be able to reproduce the (sub) sample used in this study. In this case here, that does not seem feasible. Therefore, it is strongly recommended that the authors clearly and transparently describe their methodology with complete sentences either within the main text or in the supplementary materials.

Reviewer #2: It’s disappointing that guidelines that need to be followed in journal submission are not followed even when pointed out priory. There is no line numbering in the PDF version that I received. It would be really convenient to provide suggestions. In the discussion section in page no. 7 last line you have mentioned and I quote here, ”This double burden of malnutrition is observed in almost every part of the world.” By providing references of 3 south Asian countries I don’t think you can extrapolate the findings to prevail in entire world. You need to backup such sentences with more references from various corners of world.

The major issue I find in the study is the methodology section. It needs to be more extensive. It is mentioned that it is secondary data analysis research but in page no. 3 in analysis section you have indicated that you just downloaded the estimate and their CI and further no any statistical analysis was done. Without any statistical analysis of your own and methodologies that is not transparent and that can’t be followed by future researchers to replicate similar studies in different regions or settings, I find very little significance of this study .

7. PLOS authors have the option to publish the peer review history of their article (what does this mean?). If published, this will include your full peer review and any attached files.

Reviewer #1: No

Reviewer #2: No

---

## [Author Response · Author response to Decision Letter 1]

28 Jul 2022

Reviewer #1

There are major issues with this manuscript as at present stage which has to be addressed.

1. Methodology: Transparency and thoroughness are essential in reporting, since it is strongly advised to not leave readers in the shadows about the procedure. All we know that author(s) have used secondary data from GBD database. However, things is GBD database contains large data with various information. The concern is how did you sample the required data, and how did you perform all these analyses? Therefore, sampling strategy, robust statistical analysis, software used for data analysis needs to be reported in a way that another research with access to the same database would be able to reproduce the (sub) sample used in this study. In this case here, that does not seem feasible. Therefore, it is strongly recommended that the authors clearly and transparently describe their methodology with complete sentences either within the main text or in the supplementary materials.

Response: We have attached a supplement with step-by-step guidance for any future researchers interested to replicate this study. Hope that satisfies the reviewer’s concern on transparency. 

Reviewer #2

It’s disappointing that guidelines that need to be followed in journal submission are not followed even when pointed out priory. There is no line numbering in the PDF version that I received. It would be really convenient to provide suggestions. In the discussion section in page no. 7 last line you have mentioned and I quote here, ”This double burden of malnutrition is observed in almost every part of the world.” By providing references of 3 south Asian countries I don’t think you can extrapolate the findings to prevail in entire world. You need to backup such sentences with more references from various corners of world.

The major issue I find in the study is the methodology section. It needs to be more extensive. It is mentioned that it is secondary data analysis research but in page no. 3 in analysis section you have indicated that you just downloaded the estimate and their CI and further no any statistical analysis was done. Without any statistical analysis of your own and methodologies that is not transparent and that can’t be followed by future researchers to replicate similar studies in different regions or settings, I find very little significance of this study.

Response: Sorry that the pdf version you received didn’t show line numbers. We were diligent not to miss it this time. Hope the current revised version you received appeared with line numbers. 

• We agree that the sentence extrapolating to the “world” is an over-reach with references from South Asia only. So we have removed it, and the paragraph now talks about South Asia only. 

• We however respectfully disagree with reviewer #2’s sentiment that by using GBD estimates our study has “very little significance”. Yes, GBD data are already analyzed estimates from a third party (IHME). But, numbers alone don’t tell the story, thus GBD database is widely used by researchers worldwide to answer relevant research questions on their topic, country, and region of interest, and numerous publications have come out of those endeavors. As chronic disease researchers in Nepal, we see double malnutrition and its trend to be a highly relevant and important topic to look at with a very reliable data source and discuss the relevance to our country. We have additional faith in the GBD database for use in Nepal because of the collaboration with Nepal Health Research Council (http://nhrc.gov.np/projects/estimating-burden-of-disease-for-nepal-using-globally-used-methods/). We strongly believe that by writing this paper with relevant background, presenting data in useful tables and visuals, and discussing the relevance of findings for Nepal, we have provided a significant scientific contribution in the field.

• To address the reviewer’s comment about methodology transparency, we have added a tab in the supplement file named “steps”, and mentioned step by step procedure to download for “future researchers to replicate similar studies in different regions or settings”. We also added a sentence about that supplement in the main text also. We hope this satisfies the reviewer’s concern

---

## [Editor Report · Decision Letter 2]

10 Aug 2022

Double Burden of Malnutrition in Nepal: A trend analysis of Protein-Energy Malnutrition and High Body Mass Index using the data from Global Burden of Disease 2010-2019

PONE-D-21-41047R2

Dear Dr. Pradhananga,

We’re pleased to inform you that your manuscript has been judged scientifically suitable for publication and will be formally accepted for publication once it meets all outstanding technical requirements.

Kind regards,

Pranil Man Singh Pradhan, M.D.

Academic Editor

PLOS ONE
---

## [Editor Report · Acceptance letter]

21 Sep 2022

PONE-D-21-41047R2 

Double Burden of Malnutrition in Nepal: A trend analysis of Protein-Energy Malnutrition and High Body Mass Index using the data from Global Burden of Disease 2010-2019 

Dear Dr. Pradhananga:

I'm pleased to inform you that your manuscript has been deemed suitable for publication in PLOS ONE. Congratulations! Your manuscript is now with our production department. 

Kind regards, 

on behalf of

Dr. Pranil Man Singh Pradhan 

Academic Editor

PLOS ONE